# Patient Active Approaches in Osteopathic Practice: A Scoping Review

**DOI:** 10.3390/healthcare10030524

**Published:** 2022-03-14

**Authors:** Christian Lunghi, Francesca Baroni, Andrea Amodio, Giacomo Consorti, Marco Tramontano, Torsten Liem

**Affiliations:** 1Clinical-Based Human Research Department, Foundation COME Collaboration, 65100 Pescara, Italy; clunghi@comecollaboration.org (C.L.); giacomo.consorti@gmail.com (G.C.); 2Research Department, Osteopathie Schule Deutschland, 22297 Hamburg, Germany; tliem@osteopathie-schule.de; 3Malta ICOM Education, GZR 1071 Santa Venera, Malta; andrea.amodio@icomedicine.com; 4Education Department of Osteopathy, Istituto Superiore di Osteopatia, 20126 Milan, Italy; 5Centre Pour l’Etude, la Recherche et la Diffusion Osteopathiques, 00199 Rome, Italy; m.tramontano@hsantalucia.it; 6Fondazione Santa Lucia IRCCS, 00179 Rome, Italy

**Keywords:** manipulation, osteopathic, exercise movement techniques, exercise therapy, mind-body therapies, bodyworks

## Abstract

Background: In the field of manual therapies there is a growing interest in moving from passive hands-on approaches to patient active approaches. In the osteopathic field there are both active and passive methods described as integrated in the process of care. However, this prospective linkage has not been formally explored and is not well shared in the community of practice. The present review aims to appraise the relevant literature on the functioning and principles of Patient active osteopathic approaches (PAOAs) and explore a prospective model for selecting the different types of PAOA, highlighting their integration into patient management strategies. Methods: A scoping review was conducted to analyze the relevant literature on the functioning and the different principles of PAOA and to obtain a comprehensive perspective on the phenomenon. Results: The eligible articles provide insights into the mechanisms of functioning and principles of application of active approaches to be integrated with hands-on approaches. These results provide new insights into the relevance of PAOA to clinical practice. Conclusions: The proposal, emerging from the review, may promote discussions in the community of practice and provide a road map for research towards achieving an evidence-based structure for PAOA.

## 1. Introduction

Osteopathy is a whole-body patient-centered intervention mainly focused on sustaining a person’s health processes by means of touch-based approaches focused on the somatic dysfunctions (SD) [1,2,3] present in different regions of the body. The standards that define the provision of osteopathic education and health services [4,5] suggest that osteopaths must formulate a management interprofessional plan to help patients to understand the significance and potential effect of treatment. In addition, the osteopath promotes therapeutic education and encourages patients to understand the usefulness of physical activity, exercise, lifestyle, and diet from a multi-professional health care point of view [6,7].

Osteopathic health promotion and disease prevention appears more closely connected with therapeutic education exercises [6]. For the full recovery for patients, in addition to addressing the phases of repair and alleviation of symptoms [8], the treatment must also be centered on the enhancement of individual adaptability [9]. A management plan that provides active patient approaches alongside manipulative therapies is aimed at improving the quality of the program for the individual [8,9]. Passive or active mobilization techniques or dynamic movement challenges gradually applied after shared decision making with the patient can support recovery behavior from pain, physical incapacity, or movement-related anxieties [8]. The exercises in the osteopathic field are described as approaches directed towards managing SD in different domains [10,11] and directed towards personal energy management [12]. However, several studies [13,14,15] report that these approaches are still under debate in osteopathic training and practice It is not clear whether exercise and lifestyle advice are included as interprofessional practices or whether a complementary approach is within osteopathic practice. There is currently an absence of common clinical practice frameworks regarding the integration of exercise and lifestyle advice with hands-on approaches. This may lead to misunderstandings in both patients and other health professionals distorting the professional identity.

First, we appraise the relevant literature on the functioning and principles of Patient active osteopathic approaches (PAOA). Secondly, we focus on a possible prospective model for selecting the different types of PAOA, also highlighting their integration in the patient management strategies.

## 2. Materials and Methods

As described by Levac and his colleagues [16], the authors followed the following steps:

Identify the research questions;

Identify the relevant studies;

Search strategy,Select the studies.

Compile, summarize and report the results.

The authors used the Preferred Reporting Items for Systematic Reviews and Extension of Meta-Analyses for Scoping Reviews (PRISMA) checklist and explanations to report the selected findings [17] (Figure 1).

### 2.1. Identifying the Research Questions

This scoping review explored the nature and extent of the published literature describing the functioning and principles of PAOA and their integration in patient management strategies.

### 2.2. Identifying the Relevant Studies

#### 2.2.1. Search Strategy

Two of the researchers (CL, FB) searched databases, including PUBMED, SCIENCE DIRECT, and the Cochrane Library (Table 1). The authors developed several research strings using the following keywords: manipulation, osteopathic; exercise movement techniques; exercise therapy; mind-body therapies; bodyworks; mindfulness; meditation; fascia; musculoskeletal manipulations. Search terms were modified for each database, and appropriate subheadings were used for each database searched. The search strategy included reviews, clinical trials, and observational studies. No limits of population and study outcome were applied. The search was limited to papers published in English. The authors also searched the reference lists of the articles, performing a snowball procedure.

#### 2.2.2. Select Studies

After an initial screening and deletion of irrelevant studies, three of the researchers (CL, FB, MT) selected the studies to be included for the analysis (Figure 1). To draw up the review, we developed the main theme and grouped the included items into subthemes. Articles were screened for inclusion, and those not written in English were excluded.

### 2.3. Collecting, Summarizing, and Reporting Results

Collectively, all authors discussed the chartered data to identify trends, breadth, and gaps in the literature. The results are narratively summarized.

## 3. Results

In the following paragraphs, a total of 16 articles are included that address main theme 1. Patient Active Osteopathic Approaches. Principles of applications and mechanisms of functioning. The findings are reported and grouped on the basis of their pertinence to four subthemes: two articles were included in subtheme 1.1. Fascia-oriented training; five articles were included in subtheme 1.2. Integrated mental imagery and work-in exercises; seven articles were included in subtheme 1.3. Mindfulness-based exercise; and two articles were included in subtheme 1.4. Gamification and problem-solving in the inter-enactive dyadic approach.

### 3.1. Fascia-Oriented Active Approach

Schleip and Muller [18] described the basic principles of the fascia-oriented active approach, which are proposed to be integrated into PAOA [9]. Sedentary lifestyle has been shown to cause the production of a network of multi-directional fibers, and to minimize the development of “crimps”, i.e., physical and electrical connections. Applying proper exercise leads to an improvement in connectivity in the organization of the tissue architecture. The central principles of PAOA oriented to the fascia are fascial remodeling, fascial recoil, dynamic stretching and fascial perception.

Fascial remodeling is the response of the arrangement’s network collagen fibers following a specific mechanical stimulus. Fascial active training, performed 1–2 times per week for 6–24 months, is able to affect this substitution [18].

Fascial recoil is the mechanism of elastic tissue return, stimulated through targeted active exercises, in which a preparatory phase increases the elastic tension of the fascial system, followed by a stage where the body releases the weight like a catapult [18]. Dynamic stretching oriented to the fascia consists of specific stretching performed regularly and long term in order to make the framework of the connective tissue more elastic [18]. Fascial perception consists of small and/or complex gestures used to increase awareness of perceptually ignored areas of the body, i.e., those that are related to interoceptive–proprioceptive sensory alterations [18].

### 3.2. Integrated Mental Imagery and Work-In Exercise

The musculoskeletal system represents an interoceptive body image and awareness generator. For this reason, it is preferable to use a tailored educative and cognitive approach, integrating hands-on treatment and experiential bodywork [19]. Mental imagery has been shown to have a beneficial effect on motor and cognitive output and other behavioral effects and to cause brain activity similar to what occurs with movement [20]. Mental imagery is a training method used in both prevention and rehabilitation. It can be used in combination with manual therapy or active exercise techniques to manage pain and enhance motor abilities, such as motor neuron stimulation and non-motor performance aspects, including for treatment of self-confidence and anxiety [20].

Mental imagery has already been integrated with osteopathic approaches and experiential bodywork [21,22]. For example, fascial unwinding, in which the operator starts activating slow, painless motion in the patient’s body using their hands with simultaneous use of mental imagery.

The patient follows their emergent movement, assisted by the practitioner, as the body is perceived as being unwound by ideomotor movement [21]; automatic movements are expressions of predominant ideas. Ideomotion is instinctive and unconsciously motivated activity or behavior, including excitomotor and sensorimotor actions [22].

The practitioner tries to improve interoception, including the perception of a myofascial contraction to be released and of a motor activity to be executed.

Osteopaths can do this by providing limited resistance to the patient’s action—not enough to stop it, but still enough to postpone and refine its expression. The restricted barrier of articulation or the stiffness of the soft tissue might be perceived by the patient due to the active and gentle “manual transfer” prompted by the placement of the practitioner’s hand in the regions involved [22]. In combination with integrated mental imagery or provided alone, the work-in mentality could be another complementary method to support the parasympathetic functions (rest/digestion/repair) of patients [23]. This consists of exercises performed following a breathing pattern with an oscillating harmonic rhythm. During the exercises, the attention is focused on the inherent feedback received by focusing on the amplified moving pattern of a body region or of the entire framework.

### 3.3. Mindfulness-Based Exercise

A recent review of functional magnetic resonance imaging research offers insight on the impact of combined hands-on therapy and mindfulness-based approaches on central sensitization and interoceptive shortfall conditions [24]. The interoceptive paradigm for osteopathy [25] reinforces the need for new approaches guided by practitioner [26] and patient mindfulness [27,28]. Mindfulness is often used as an umbrella term to characterize many practices, processes, and characteristics [24]. These focus on the non-directed process of managing attention to present experientail perceptions, such as thoughts, emotions, and sensations [29]. Different osteopathic approaches [28], such as body scan mindfulness practice, involve sequentially directing attention to various parts of the body, without judgment, to develop the regulation of somato-visceral perceptions, sensations, and cognitive judgments [28,29]. Moreover, the use of oscillation, vibration, and spontaneous myofascial and neurogenic tremors, both in osteopathy [30] and mindfulness-based practice [29], are today promoted to release stress and restore the body’s homeostasis.

### 3.4. Gamification and Problem-Solving in the Inter-Enactive Dyadic Approach

Gamification is an example of developing a team-spirit patient problem-solving atmosphere, where challenges can be discussed and improved in a task-oriented way [31]. Performing the active assistive exercises in a peaceful and fun way and using metaphors to describe the routine allows the information to become sub-cortical. PAOA is mainly partner-based, emphasizing novel environmental constraints, safety, and fun. During the execution of the exercises, the attention is focused on movement rather than activating the individual muscle. The participant’s body–brain unity is able to re-learn adaptation processes through exposure to new or challenging situations in a safe environment [31]. As argued by enactivists, embodied humans improve their sense-making by action-oriented relational strategies distributed throughout the brain–body–environment [32].

The enactive model describes a human sensory action cycle that drives structure and function adaptation to better match the environment [32]. The exteroceptive, proprioceptive, and interoceptive sensory systems support the brain in generating predictions about its external and internal environment and adaptive actions in order to maintain health [32]. The failure to process and integrate multisensory bodily signals followed by prediction errors may have relevance for the physiological regulatory process in health and disease [32].

## 4. Discussion

The results presented in the present review indicate the ways in which PAOA can integrate hands-on treatment with different motor, cognitive and behavioral strategies.

These complementary approaches seem to activate neuromyofascial and tissue remodeling, modify body image and awareness, and can help in the management of the stress levels.

A putative model for PAOA in order to make clinical practical use of the results of this review is presented in Appendix A, Table A1.

The integration of hands-on treatment with mental imagery, mindfulness, and exercises is supposed to improve the functionality of body systems [9,20,27,28]. It is presumed that this will have an effect on the circulatory system by stimulating the production or retention of lubricants between fascial fibers [18]. The neurologic system, including sensory pathways, can be enabled, e.g., spinal reflexes and corticospinal excitability [19]. The enhanced neural activations improve proprioceptive signaling from the mechanoreceptors embedded in the fascial system [19]. Directing the patient’s attention towards the touched or moved areas promotes body awareness, identification, body image, and body schema, favoring psychological responsiveness. All of the above-mentioned approaches can be employed using gamification attitudes for the facilitation of problem solving in the participant/osteopath/environment triadic relationship with respect to the patient’s illness.

During PAOA administration, the osteopath should use some education strategies to tailor the sessions. For example, pain physiology education could be implemented to reconceptualize pain [33], and advice about healthy lifestyles, including diet, could be provided to ameliorate the patient’s negative quality of life and illness outcomes [34]. This can be regarded as a component of the osteopath’s explanation of treatment rationale when discussing treatment with the patient [33]. Illustrations, examples, and metaphors are frequently used to communicate advice [33].

Educative information is delivered orally by the practitioner, while graphical summaries and visual representations are displayed on a screen or paper [33]. The food pyramid, adapted for different populations, is used as an educational tool to improve nutrition knowledge [25]. During such encounters, participants are prompted to ask questions, and their feedback can be used to customize the advice [33]. The need to integrate passive approaches for patients has often been noted in manual therapies [35], as well as in osteopathy [9]. Indeed, manual therapists have used terms and phrases in a creative way to convey prevalent, specific theories [36]. This vocabulary emphasizes the distinction of manual therapists from other health care occupations [36]. This may lead to patient confusion about health care issues and reduce interprofessional collaborations.

It could be helpful to use metaphors to explain concepts [33], but osteopaths must be aware that positional terminology (e.g., flexed and rotated vertebra) is anachronistic and potentially dangerous for patients. Adopting such terminology can establish the idea of a severe condition in the mind of an anxious person, leading to catastrophizing, fear-avoidance behavior, and dependence on passive approaches for something that “is gone out of place, and needs to be fixed” [37].

As reported in a recently published commentary [38,39] and review [40], PAOA should be regarded as a strategy for integrating passive approaches in osteopathy with more patient-centered care and improving interprofessional collaborations. PAOA could allow patients to improve daily movements that they feel unable to perform appropriately because of their symptoms (i.e., comparative signs or familiar symptoms). Natural movement is performed in a specific way, instead of in a stereotypical way, by changing environmental, strategic or operational factors. The body–brain unity learns to adapt through exposure to new or changing circumstances, particularly when there is a sense of being in a safe setting, such as playing [31].

To the best of our knowledge, the present thematic analysis is the first scoping review study conducted on the topic. The present scoping review highlights a limited amount of literature on hands-off strategies and active patient approaches commonly used in osteopathic practice. Future studies and consensus workshops conducted in the community of practice are required in order to develop a common framework for an evidence-based osteopathic practice that involves the patient as an active part of the treatment to promote wellbeing in an interdisciplinary person-centered care process.

## 5. Conclusions

The results of this review tentatively suggest the possible integration of PAOA into osteopathic practice, but the fact that there is a lack of a consensus about the description of PAOA may lead to confusion among patients and practitioners. Different professions use exercise for the management of their patients, and it could be useful to facilitate a discussion in the field of active patient osteopathic approaches. This proposal, emerging from this review, may guide a research agenda (Appendix A, Table A2) for achieving an evidence-based structure for PAOA. Additionally, defining the distinctive osteopathic practice while using active assistive exercise for the management of patients.

## Figures and Tables

**Figure 1 healthcare-10-00524-f001:**
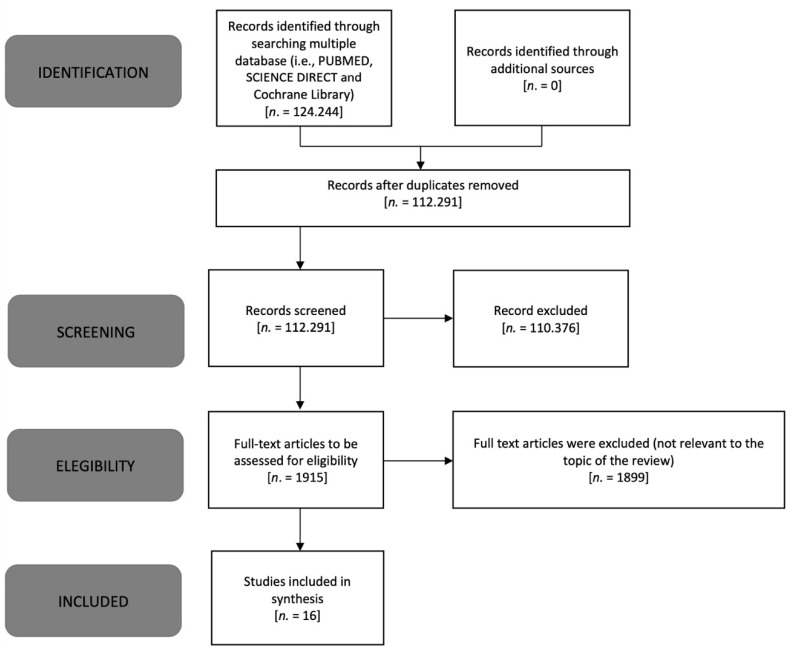
PRISMA flow diagram modified for scoping review [17].

**Table 1 healthcare-10-00524-t001:** Literature research.

**Keywords:** manipulation, osteopathic; musculoskeletal manipulations; exercise movement techniques; exercise therapy; mind-body therapies; bodyworks; mindfulness; meditation; fascia.			
**Pubmed search builder:** (((((((“Manipulation, Osteopathic” [Mesh]) OR “Musculoskeletal Manipulations” [Mesh]) AND “Exercise Movement Techniques” [Mesh]) OR “Exercise Therapy” [Mesh]) OR “Mind-Body Therapies” [Mesh]) OR “Mindfulness” [Mesh]) OR “Meditation” [Mesh]) OR “Fascia” [Mesh]			
**Items grouped with respect to theme and subthemes**			
1.Patient active osteopathic approaches. Principles of application and mechanisms of functioning (total n. = 16).			
1.1. Fascia-oriented active approach (n. = 2).	1.2. Integrated mental imagery and work-in exercise (n. = 5).	1.3. Mindfulness-based exercise (n. = 7).	1.4. Gamification and problem-solving in the inter-enactive dyadic approach (n. = 2).
Lunghi et al., 2016 [9] Schleip and Muller, 2013 [18]	Calsius et al., 2016 [19] Abraham et al., 2020 [20] Minasny, 2009 [21] Dorko 2003 [22] Wallden, 2012 [23]	Casals-Gutiérrez and Abbey, 2020 [24] D’Alessandro et al., 2016 [25] Comeaux, 2005 [26] Nanke and Abbey, 2017 [27] Liem and Neuhuber, 2020 [28] Liem and Lunghi, 2021 [29] Comeaux, 2005 [30]	Liebenson 2018 [31] Esteves et al., 2022 [32]

## Data Availability

The data presented in this study are available on request from the corresponding author.

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
