# Peer review of "Patient Active Approaches in Osteopathic Practice: A Scoping Review"

_healthcare, 2022, doi:10.3390/healthcare10030524_

Round 1

Reviewer 1 Report

Comments

Ln42 – “view.[6,7]” - Check the references regulations please and correct everywhere in the text where needed.

Ln53 – “studies[13-15]” - Space needed here. Please check everywhere in the text and correct it.

Ln65 – “colleagues16” – Is this a citation. Please recheck and write it correctly.

Ln100 – Table 1. Please consider presenting the table 1 after the PRISMA flow diagram. I suppose that first the search of papers was done, and after that inclusion of the eligible studies.

Ln102 - Prisma flow diagram needs to be corrected. I guess that “records excluded for not being relevant to the topic of the review” (n=110.376) should be moved to one step before. Please do the calculation again and correct it. I suggest making this diagram again from the start more attentively. Additionally, check is the numbering is correct: “n.” or “n=” ?

Ln151 – “practicioner” - Do you mean practitioner? Please recheck and make corrections.

Ln258 “Limitations” - Please consider placing the limitations within the discussion section.

General comments

Good work, this scoping review is informative and interesting. The mentioned approaches should certainly be further investigated and discussed.

I suggest reading it carefully and preparing the text as described in instruction of this journal. It is advisable to check the references again as well.

Author Response

We thank the Editors and Reviewers for their constructive, qualified, and pertinent comments, that we have carefully considered in this revision, as well as the opportunity to resubmit an improved version of our article.

The Reviewers' comments are in black font, while our responses are in blue.

Reviewer 1.

Ln42 – “view.[6,7]” - Check the references regulations please and correct everywhere in the text where needed.

Response: Thank you. We placed all references in square brackets [ ], and placed them before the punctuation.

Ln53 – “studies[13-15]” - Space needed here. Please check everywhere in the text and correct it.

Response: Thanks. We added space between “studies” and “[13-15]” and everywhere was needed in the manuscript.

Ln65 – “colleagues16” – Is this a citation. Please recheck and write it correctly.

Response: Thank you. We rechecked it and corrected it into “colleagues [16]”.

Ln100 – Table 1. Please consider presenting the table 1 after the PRISMA flow diagram. I suppose that first the search of papers was done, and after that inclusion of the eligible studies.

Response: Thank you for your comment. We changed the order of Table 1 and the PRISMA flow diagram as suggested.

Ln102 - Prisma flow diagram needs to be corrected. I guess that “records excluded for not being relevant to the topic of the review” (n=110.376) should be moved to one step before. Please do the calculation again and correct it. I suggest making this diagram again from the start more attentively. Additionally, check is the numbering is correct: “n.” or “n=” ?

Response: Thank you. We corrected the PRISMA flow diagram as suggested.

Ln151 – “practicioner” - Do you mean practitioner? Please recheck and make corrections.

Response: Thank you. We correct “practiocioner”  into “practitioner”

Ln258 “Limitations” - Please consider placing the limitations within the discussion section.

Response: Thanks. We paced the limitations in the discussion section.

General comments

Good work, this scoping review is informative and interesting. The mentioned approaches should certainly be further investigated and discussed.

I suggest reading it carefully and preparing the text as described in instruction of this journal. It is advisable to check the references again as well.

Thank you for your comment. We have revised the manuscript according to the journal’s instructions.

Reviewer 2.

Comments and Suggestions for Authors

This paper supports the positive effect of  Patient Active Osteopathic Approaches (PAOA) in osteopathic practice by providing evidence for its positive effect. Moreover it clearly presents the variability in specific methodologies among different published approaches. The merit of capturing the different practices lies on the ability to use it as a starting point for further discussion among related professionals in order to experiment and reach a consensus on the optimal workflow in such practices. Moreover, such a work can encourage other practitioners to share their experience, contributing thus to a more thorough overview of current practices and outcomes.

Minor comments
L 47 : consider adding with after alongside 

Response: Thank you. We added “with” after “alongside”

L 147: please rephrase to make clear assisted by the practitioner or himself

Response: Thank you. We rephrase the sentence into “assisted by the practitioner”

L 151 : consider substituting try with tries 

Response: Thank you for your comment.  We change “try” into “tries”

L 232 : on the other hand is used twice. The meaning is confusing

Response: Thank you for your comment.  We changed the sentence from “On the one hand, iIt could be helpful to use metaphors to explain concepts; [33];  on the other hand, but osteopaths must be aware that positional terminology (i.e., flexed and rotated vertebra) is anachronistic and potentially dangerous for patients. “  into  “It could be helpful to use metaphors to explain concepts [33]; but osteopaths must be aware that positional terminology (i.e., flexed and rotated vertebra) is anachronistic and potentially dangerous for patients.”

Reviewer 2 Report

This paper supports the positive effect of  Patient Active Osteopathic Approaches (PAOA) in osteopathic practice by providing evidence for its positive effect. Moreover it clearly presents the variability in specific methodologies among different published approaches. The merit of capturing the different practices lies on the ability to use it as a starting point for further discussion among related professionals in order to experiment and reach a consensus on the optimal workflow in such practices. Moreover, such a work can encourage other practitioners to share their experience, contributing thus to a more thorough overview of current practices and outcomes.
Minor comments
L 47 : consider adding with after alongside 
L 147 please rephrase to make clear assisted by the practitioner or himself
L 151 : consider substituting try with tries 
L 232 : on the other hand is used twice. The meaning is confusing

Author Response

(The authors gave the same response as above.)

Round 2

Reviewer 1 Report

Fix flow diagram!

http://www.prisma-statement.org/PRISMAStatement/FlowDiagram

Author Response

Dear Editor and we would like to thank you and the reviewers for giving us the opportunity to improve our manuscript. We ho pe that the new version of the manuscript is now suitable for the pubblication.

Reviewer Comments and Suggestions for Authors:

Fix flow diagram!

http://www.prisma-statement.org/PRISMAStatement/FlowDiagram

Response: Dear Reviewer, thank you for your suggestion, we have modified Figure 1 according to the PRISMA Flow Diagram Modified for Scoping Review.

Tricco, A.C.; Lillie, E.; Zarin, W.; O'Brien, K. K.; Colquhoun, H.; Levac, D.; Moher, D.; Peters, M.; Horsley, T.; Weeks, L.; Hempel, S.; Akl, E. A.; Chang, C.; McGowan, J.; Stewart, L.; Hartling, L.; Aldcroft, A.; Wilson, M. G.; Garritty, C.; Lewin, S.; Godfrey, C.M.; Macdonald, M.T.; Langlois, E.V.; Soares-Weiser, K.; Moriarty, J.; Clifford, T.; Tunçalp, Ö.;Straus, S. E. PRISMA Extension for Scoping Reviews (PRISMA-ScR): Checklist and Explanation. Ann. Intern. Med. 2018, 169, 7, 467-473. doi:10.7326/M18-0850
